# Charcot-Marie-Tooth 2B mutations in *rab7* cause dosage-dependent neurodegeneration due to partial loss of function

Smita Cherry[1†], Eugene Jennifer Jin[1†], Mehmet Neset Özel[1], Zhiyuan Lu[2], Egemen Agi[1], Dong Wang[1], Wei-Hung Jung[3], Daniel Epstein[1], Ian A Meinertzhagen[2], Chih-Chiang Chan[1,3]*, P Robin Hiesinger[1,4]*

[1]Department of Physiology, University of Texas Southwestern Medical Center, Dallas, United States; [2]Department of Psychology and Neuroscience, Dalhousie University, Halifax, Canada; [3]Department of Physiology, National Taiwan University, Taipei, Taiwan; [4]Green Center for Systems Biology, University of Texas Southwestern Medical Center, Dallas, United States

**Abstract** The small GTPase Rab7 is a key regulator of endosomal maturation in eukaryotic cells. Mutations in *rab7* are thought to cause the dominant neuropathy Charcot-Marie-Tooth 2B (CMT2B) by a gain-of-function mechanism. Here we show that loss of *rab7*, but not overexpression of *rab7* CMT2B mutants, causes adult-onset neurodegeneration in a *Drosophila* model. All CMT2B mutant proteins retain 10–50% function based on quantitative imaging, electrophysiology, and rescue experiments in sensory and motor neurons in vivo. Consequently, expression of CMT2B mutants at levels between 0.5 and 10-fold their endogenous levels fully rescues the neuropathy-like phenotypes of the *rab7* mutant. Live imaging reveals that CMT2B proteins are inefficiently recruited to endosomes, but do not impair endosomal maturation. These findings are not consistent with a gain-of-function mechanism. Instead, they indicate a dosage-dependent sensitivity of neurons to *rab7*-dependent degradation. Our results suggest a therapeutic approach opposite to the currently proposed reduction of mutant protein function.

*For correspondence: chancc1@ntu.edu.tw (C-CC); robin.hiesinger@utsouthwestern.edu (PRH)

†These authors contributed equally to this work

Competing interests: The authors declare that no competing interests exist.

## Introduction

Several neuropathies, lysosomal storage diseases and neurodegenerative disorders primarily affect the nervous system, despite underlying defects in cellular processes that occur in all cells (*Schultz et al., 2011*; *Wang et al., 2012*). Charcot-Marie-Tooth 2B (CMT2B) is a sensory neuropathy that primarily affects some of the longest axon projections in the human body and is caused by mutations in the *rab7* locus. *rab7* encodes a GTPase that regulates endolysosomal degradation in all cells (*Elliott et al., 1997*). All known mutations in CMT2B patients alter highly conserved amino acids in Rab7 and cause pathology in heterozygosity (*Kwon et al., 1995*; *Verhoeven et al., 2003*; *Houlden et al., 2004*; *Meggouh et al., 2006*). Hence, CMT2B is a genetically dominant disease.

Several studies have proposed a neuron-specific gain-of-function mechanism of the CMT2B alleles to explain the dominant neuronal phenotype of this ubiquitous gene (*Spinosa et al., 2008*; *Cogli et al., 2010*; *McCray et al., 2010*; *Cogli et al., 2013*; *Zhang et al., 2013*). In support of this hypothesis, several dominant functions of CMT2B mutant Rab7 have been described based on overexpression of the mutant proteins in neuronal or non-neuronal cultured cells. For example, CMT2B protein expression leads to altered EGF degradation in HeLa cells (*Spinosa et al., 2008*), decreased upregulation of the

**eLife digest** Charcot-Marie-Tooth disease is an inherited disorder of the nervous system with symptoms that typically begin in adolescence or early adulthood. The sensory and motor nerves gradually degenerate, causing muscles to waste away and leading to the loss of touch sensation across the body. One subtype of the disease—Charcot-Marie-Tooth 2B—is caused by mutations in a gene called *rab7*, which codes for a protein that helps to regulate the breakdown of waste proteins inside cells.

Charcot-Marie-Tooth 2B is described as a genetically dominant disorder because all patients have one wild type copy and one mutant copy of the *rab7* gene. Overexpression of the mutant gene in cells grown in culture alters many of the signaling pathways inside the cells, but it is unclear whether these alterations cause the pathology seen in the disease.

Now, Cherry et al. have obtained new insights into the genetics of Charcot-Marie-Tooth 2B by creating the first animal model of the disorder. Fruit flies that did not have the *rab7* gene in the light-sensitive sensory neurons in their eyes were used to compare normal and mutant cells. While the two cell types were initially similar, the mutant cells gradually degenerated in the adult animal. By contrast, cells that overexpressed a mutant form of the *rab7* gene continued to function normally throughout adulthood. Moreover, when mutant Rab7 proteins were introduced into the cells that lacked the *rab7* gene, the proteins restored the cells' sensitivity to light. These results suggest that mutant Rab7 proteins do not cause degeneration; instead, it is the loss of normal Rab7 function that causes problems.

At present, most research into treatment is aimed at finding ways to reduce the activity of mutant Rab7 proteins. However, the work of Cherry et al. suggests that increasing the activity of normal Rab7 proteins—or increasing the activity of alternative pathways that degrade waste proteins—may help to restore nerve function in this, and possibly other, neurodegenerative diseases.

growth-associated protein 43 in PC12 cells (*Cogli et al., 2010*), increased interaction with the filament protein peripherin in Neura2A cells (*Cogli et al., 2013*), modulatory effects on JNK signaling in N1E-115 cells (*Yamauchi et al., 2010*), accumulation of the NGF receptor TrkA in cultured dorsal root ganglia cells (*Zhang et al., 2013*), and altered EGF receptor signaling in HeLa, BHK-21 and A431 cells (*Basuray et al., 2013*), amongst others. Furthermore, a recent report has suggested that overexpression of CMT2B mutants in HeLa and PC12 cells dominantly reduces *rab7* function (*Basuray et al., 2013*). It is unclear which ones of these effects are causally linked to the neuropathy in aging sensory and motor neurons in humans. Since Rab7 is a key protein required for endolysosomal function in all cells, its loss or gain-of-function is predicted to directly or indirectly affect many signaling pathways over time. In addition, it is currently unclear whether overexpression of the CMT2B mutant proteins actually causes axon terminal degeneration in a sensory or motor neuron. Indeed, overexpression in at least one cell culture system revealed no obvious toxic effects (*McCray et al., 2010*). Consequently, the mechanism underlying the genetic dominance and the putative gain-of-function underlying the pathology of CMT2B remains unclear.

Rab7 has a well understood and critical role in converting Rab5-positive early endosomes into late endosomal compartments and thereby represents a key step in endolysosomal maturation in all cells (*Bucci et al., 2000*; *Rink et al., 2005*; *Poteryaev et al., 2010*). Rab7 GTPase function is biochemically well defined and GTP-locked 'constitutively active' and GDP-locked 'dominant negative' mutants have been tested and utilized in a plethora of systems, including human cell lines and *Drosophila* (*Mukhopadhyay et al., 1997a*, *1997b*; *Bucci et al., 2000*; *Zhang et al., 2007*). More recently, extensive biochemical characterizations of all four CMT2B proteins revealed that the majority of the overexpressed protein is in a GTP-bound form when compared to the GDP-bound form (*Spinosa et al., 2008*). However, the same study showed that the overall binding to both GTP and GDP is drastically reduced for the CMT2B proteins, unlike a constitutively active form (*Spinosa et al., 2008*). In addition, a 2.8 Å crystal structure of one of the CMT2B proteins revealed no intrinsic GTPase defect (*McCray et al., 2010*). Furthermore, the CMT2B variants can at least partially rescue defects caused by reduced *rab7* levels (*Spinosa et al., 2008*; *McCray et al., 2010*). In summary, neither overexpression studies in cell culture

nor comprehensive biochemical analyses have so far pinpointed a molecular mechanism that directly causes adult-onset loss of synaptic function in sensory and motor neurons. In mouse (*Kawamura et al., 2012*) and *Caenorhabditis elegans* (*Kinchen et al., 2008*; *Skorobogata and Rocheleau, 2012*) *rab7* null mutants cause embryonic lethality, which has so far precluded the analysis of neuronal phenotypes. In this study, we used the fruit fly *Drosophila melanogaster* to analyze the role and mechanism of CMT2B mutant Rab7 proteins in sensory and motor neurons in vivo. Our findings did not uncover a neuron-specific dominant gain- or loss-of-function for the CMT2B alleles. Instead, our findings indicate that the CMT2B alleles are partial loss of function alleles of *rab7* that cause, in a dosage-dependent manner, first synaptic and subsequently neuronal degeneration.

## Results

### Loss of *rab7* causes adult-onset loss of synaptic function and subsequent neurodegeneration in *Drosophila* photoreceptor sensory neurons

Our recent functional profiling of all *rab* GTPases in *Drosophila* suggested an increased neuronal demand for *rab7* function compared to other cell types based on elevated neuronal expression of *rab7*, but not *rab5* or *rab11* (*Chan et al., 2011*). To investigate an enhanced or specialized role for *rab7* in the nervous system, we generated a null mutant by replacing the *rab7* open reading frame with a Gal4 knock-in cassette (*Figure 1A*); the Gal4 knock-in provides a means to express any gene of interest under the endogenous regulatory elements of *rab7* in either heterozygous or homozygous *rab7* mutants (*Brand and Perrimon, 1993*; *Chan et al., 2011*, *2012*). Loss of *rab7* leads to lethality between 50–80% of pupal development (P+50%–P+80%) with no gross morphological abnormalities. Loss of the maternal contribution causes lethality in fully developed embryos with no obvious developmental defects. Wholemount preparations of the brain in null mutant pupae at P+35% reveal a loss of Rab7 immunolabeling to background levels (*Figure 1B,C*) (*Chinchore et al., 2009*; *Chan et al., 2011*). By P+50% substantial accumulations of the endosomal marker Hrs become apparent, although the overall brain structure appears normal (*Figure 1D,E*). In addition, immunolabeling of photoreceptor axons indicates that cell-type specification and axon pathfinding are normal, but the photoreceptor membrane protein Chaoptin (24B10) accumulates in the brain (*Figure 1F,G*). Overexpressing UAS-YFP-Rab7 under control of the *rab7*^Gal4-knock-in in the homozygous mutant rescues these phenotypes (*Figure 1F–H*).

To study the effect of loss of *rab7* on sensory neuronal survival, we generated genetic mosaics with mutant photoreceptor neurons in otherwise heterozygous chimeric flies (*Newsome et al., 2000*; *Chotard et al., 2005*; *Mehta et al., 2005*). Photoreceptors are sensory neurons of the peripheral nervous system. Since blind flies are viable under laboratory conditions, photoreceptor neurons provide a model for the in vivo study of mutations that would be lethal if present in other tissues. Eyes mutant for *rab7* look indistinguishable from wild-type, suggesting that *rab7* mutant photoreceptors develop normally (*Figure 1I,J*). Measurements of photoreceptor function using electroretinogram (ERG) recordings (*Coombe, 1986*) revealed no defects in the amplitudes of light-evoked responses (*Figure 1K,L*) and synaptic function (as indicated by the 'on' transient; *Figure 1K,M*) in newly hatched adults. We conclude that *rab7* mutant photoreceptor neurons develop and initially function without obvious defects. In contrast, 5-day-old *rab7* mutants that were raised in constant ambient light (~600 Lux, see 'Materials and methods') exhibit almost complete loss of synaptic function (*Figure 1M*). Under the same conditions, wild-type photoreceptors exhibit no significant reduction in synaptic function. By contrast, the photoreceptor response amplitude does not significantly differ between wild-type and *rab7*, suggesting that synaptic function is more sensitive than cell body function to loss of *rab7*. Furthermore, the age-dependent synaptic defect can be fully rescued by minimizing their stimulation through raising the flies in the dark (*Figure 1M*, *Figure 1—figure supplement 1A*). Similarly, electron microscopy of *rab7* mutant photoreceptors reveals increased degeneration of cellular structure after 5 days in constant light compared with wild-type (*Figure 1N*), while 5-day dark-reared *rab7* mutants and controls exhibit no obvious defects (*Figure 1—figure supplement 1B,C*). Synaptic terminals of light-exposed *rab7* photoreceptors exhibit large-scale degeneration that is absent in unstimulated *rab7* mutant terminals (*Figure 1O,P*). These findings suggest that in the absence of neuronal stimulation *rab7* in photoreceptor neurons is partially dispensable at least in young flies. Overall, our analysis of *rab7*-deficient sensory neurons revealed progressive defects that appear first as a loss of synaptic function but lead ultimately to degeneration of the entire neuron.

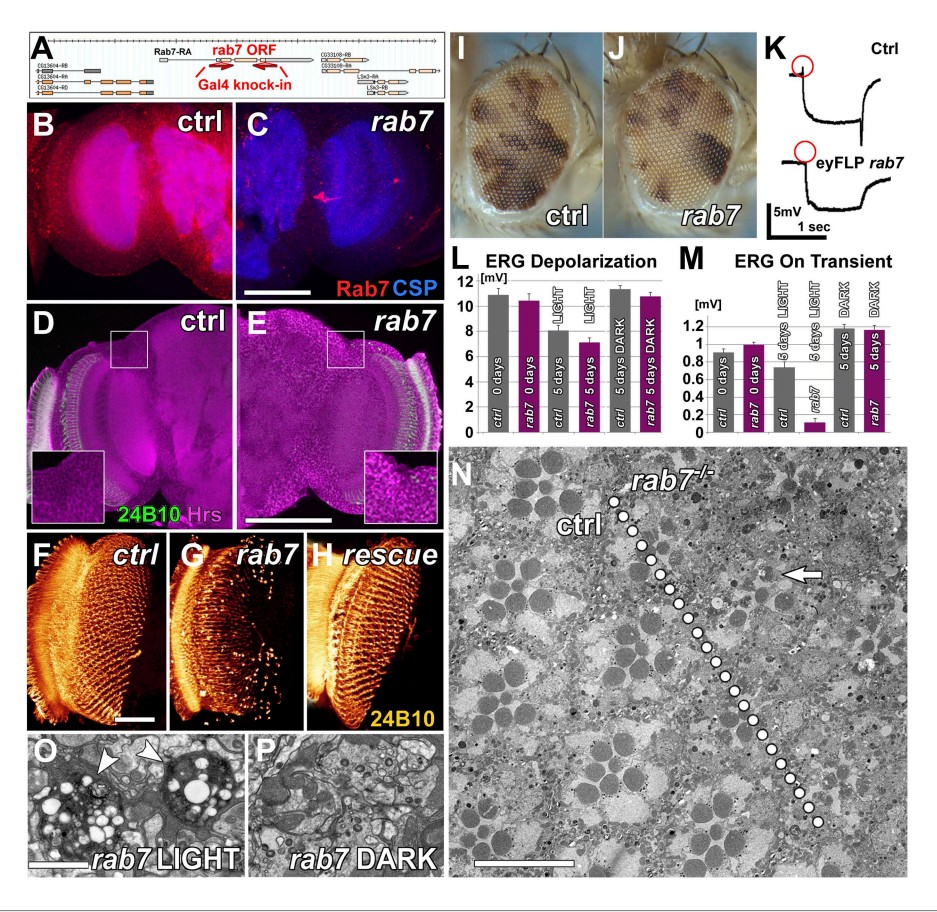

**Figure 1**. Loss of *rab7* in neurons causes adult-onset degeneration that begins with a loss of synaptic function. (**A**) Knock-out strategy: replacement of the complete *rab7* open reading frame with a Gal4 knock-in cassette (**Chan et al., 2011, 2012**). (**B and C**) Pupal brains at P+35% for wild-type (**B**) and the *rab7* mutant (**C**). Red: Rab7, Blue: synaptic vesicle marker CSP. Note that the red labeling in the center of (**C**) stems from 3xP3-RFP expression that marks the knock-in cassette. (**D and E**) Pupal brain at P+50% from wild-type (**D**) and the *rab7* mutant (**E**). Green: photoreceptor-specific mAb 24B10; magenta: the endosome marker HRS. (**F–H**) 3D visualization of photoreceptor axon projections in ctrl, *rab7* homozygous mutant, and a *rab7* homozygous mutant expressing UAS-YFP-Rab7 (rescue) (**Zhang et al., 2007**). (**I and J**) Genetic mosaics with *rab7* mutant photoreceptors in otherwise heterozygous flies exhibit no eye development defects. (**K–M**) Electroretinogram (ERG) responses from flies with *rab7* mutant eyes. Light stimulation for 5 days leads to the almost complete loss of synaptic function (ERG 'on' transient, **M**); despite normal photoreceptor responses to light (ERG Depolarization, **L**). (**K**) Sample ERG traces from 5-day old flies. (**N–P**) Electron microscopy of mutant eyes showing rhabdomere degeneration in *rab7* mutant clones (arrow) (**N**) and synaptic terminals (**O and P**). Note that the presence of pigment between ommatidia marks patches of wild-type ommatidia (compare **I and J** and arrowheads in *Figure 1—figure supplement 1C*). (**O**) Light stimulation leads to vacuolarization and degeneration of *rab7* synaptic terminals (arrowheads). Scale bar in (**C**) for (**B and C**) and (**E**) for (**D and E**): 50 µm; in (**F**) for (**F–H**): 20 µm; in (**N**): 10 µm; in (**O**) for (**O and P**): 1 µm.

The following figure supplements are available for figure 1:

**Figure supplement 1**. Functional and morphological degeneration in *rab7* mutant photoreceptors in 5-day light or 5-day dark-raised flies.

## Charcot-Marie-Tooth 2B mutations affect Rab7 protein localization, but have no dominant effect on motor neuron function

Adult-onset synaptic degeneration is a hallmark of many human sensory neuropathies. Indeed, four independent mutations in *rab7* that cause the sensory neuropathy CMT2B in patients have been characterized (**Verhoeven et al., 2003**; **Houlden et al., 2004**; **Meggouh et al., 2006**). The four amino acids

altered by these mutations (L129, K157, N161 and V162) are 100% conserved in the single *Drosophila rab7* ortholog, including their precise location in the primary sequence of identical length (*Figure 2A*). All four mutations cause the CMT2B neuropathy independently and dominantly in heterozygote patients, which has led to a focus on identifying a putative gain-of-function effect of the disease mutants (*Spinosa et al., 2008*; *McCray et al., 2010*; *Cogli et al., 2013*; *Zhang et al., 2013*). Furthermore, it has been proposed that the disease mutations might mimic the constitutively active *rab7^Q67L*, a known gain-of-function mutation (*Mukhopadhyay et al., 1997b*; *De Luca et al., 2008*; *Spinosa et al., 2008*). To test the gain-of-function hypothesis in vivo, we generated transgenic flies for the expression of all four CMT2B mutants. *rab7^WT*, *rab7^Q67L* and the well-characterized dominant negative *rab7^T22N* variants have previously been generated and successfully used in *Drosophila*, but their random genomic integrations preclude comparable levels of expression (*Zhang et al., 2007*). To compare these variants quantitatively with the CMT2B mutations at identical expression levels, we generated new transgenic lines for *rab7^Q67L*, *rab7^T22N* and *rab7^WT* using the same insertion site in the genome. Finally, we generated two transgenic fly lines for the expression of the human *rab7* ortholog *rab7A*, one for wild-type *hrab7A* and one of the CMT2B mutation K157N. All nine fly and human *rab7* variants (WT, Q67L, T22N, K157N, L129F, N161T, V162M, hrab7A-WT, hrab7A-K157N) were N-terminally tagged with the YFP variant Venus, similar to previous experiments showing that N-terminal tagging does not interfere with *rab* GTPase function (*Zhang et al., 2007*).

The gain-of-function hypothesis predicts that overexpressing the constitutively active (GTP-bound) Rab7-Q67L or the CMT2B variants should cause a disease-related phenotype. We expressed all nine variants under the endogenous *rab7* regulatory elements using the *rab7^Gal4-knock-in* in heterozygotes. The *rab7^Gal4-knock-in* -driven UAS-venus-*rab7* expresses all variants in the endogenous spatiotemporal expression pattern of *rab7*. However, potential caveats include (1) that the *rab7^Gal4-knock-in* may exhibit small differences from the endogenous pattern not detected here, (2) that the UAS transgenes may exhibit small differences in expression levels even though they are inserted in the same landing site-sand in identical genetic background, and (3) that the Gal4/UAS amplification can increase overall protein levels (*Brand and Perrimon, 1993*; *Chan et al., 2011*). Western Blot analysis of total protein extract from fly heads revealed that all Venus-Rab7 proteins are mildly overexpressed at ~1.8 to 3-fold of the levels of the endogenous Rab7 protein (*Figure 2—figure supplement 1*). We established nine transgenic lines for each of the nine UAS-venus-*rab7* transgenes in flies heterozygous for the *rab7^Gal4-knock-in* null allele (see "Generation of nine UAS-venus-*rab7* constructs and transgenic lines"). Hence, each line stably expresses one of the nine mutant *rab7* variants under the endogenous regulatory elements of *rab7* with around twofold protein levels of the endogenous protein in a background of 50% of the endogenous protein. Remarkably, all nine lines are viable as adults without obvious behavioral defects or altered lifespan.

To measure whether loss of *rab7* or expression of the CMT2B variants causes functional defects in motor neurons, we performed electrophysiological recordings at the larval neuromuscular junction of the *rab7* null mutant and for overexpression of three *Drosophila* and human CMT2B variants. If the CMT2B variants exert a dominant effect on synaptic function, these recordings should reveal an overexpression defect even if the null mutant does not. As shown in *Figure 2B–D*, both the frequency and amplitudes of spontaneous single vesicle release events are indistinguishable between control, *rab7* null mutant and CMT2B protein overexpression. Similarly, evoked neurotransmission exhibits no defects for any genotype (*Figure 2E,F*). The absence of synaptic defects in the *rab7* null mutant is consistent with the electroretinogram recordings of photoreceptor function in young adults shown above (*Figure 1K–M*). In addition, these findings indicate that mild overexpression of CMT2B mutant proteins in their endogenous expression pattern in an entire animal in vivo does not obviously affect synaptic function in motor neurons.

To compare the behavior and subcellular distribution of all mutant Rab7 proteins, we investigated the subcellular location of the nine *Drosophila* and human Venus-tagged proteins. Venus-Rab7-WT-positive compartments are present in both presynaptic boutons and the surrounding muscle (*Figure 2G*). Expressing the constitutively active Rab7-Q67L leads to increased compartment numbers mostly in the surrounding muscle, while Rab7-T22N exhibits only diffuse labeling (*Figure 2G*). In contrast, the CMT2B variants accumulate in the center of synaptic boutons (arrowheads in *Figure 2G*). Functional Rab7 is characterized by dynamic colocalization first with the early endosomal marker Rab5 and second with the early-to-late endosomal ESCRT protein Hrs (*Lloyd et al., 2002*). Colocalization analysis with these markers therefore provides a means to assess the functional state of the nine Venus-tagged

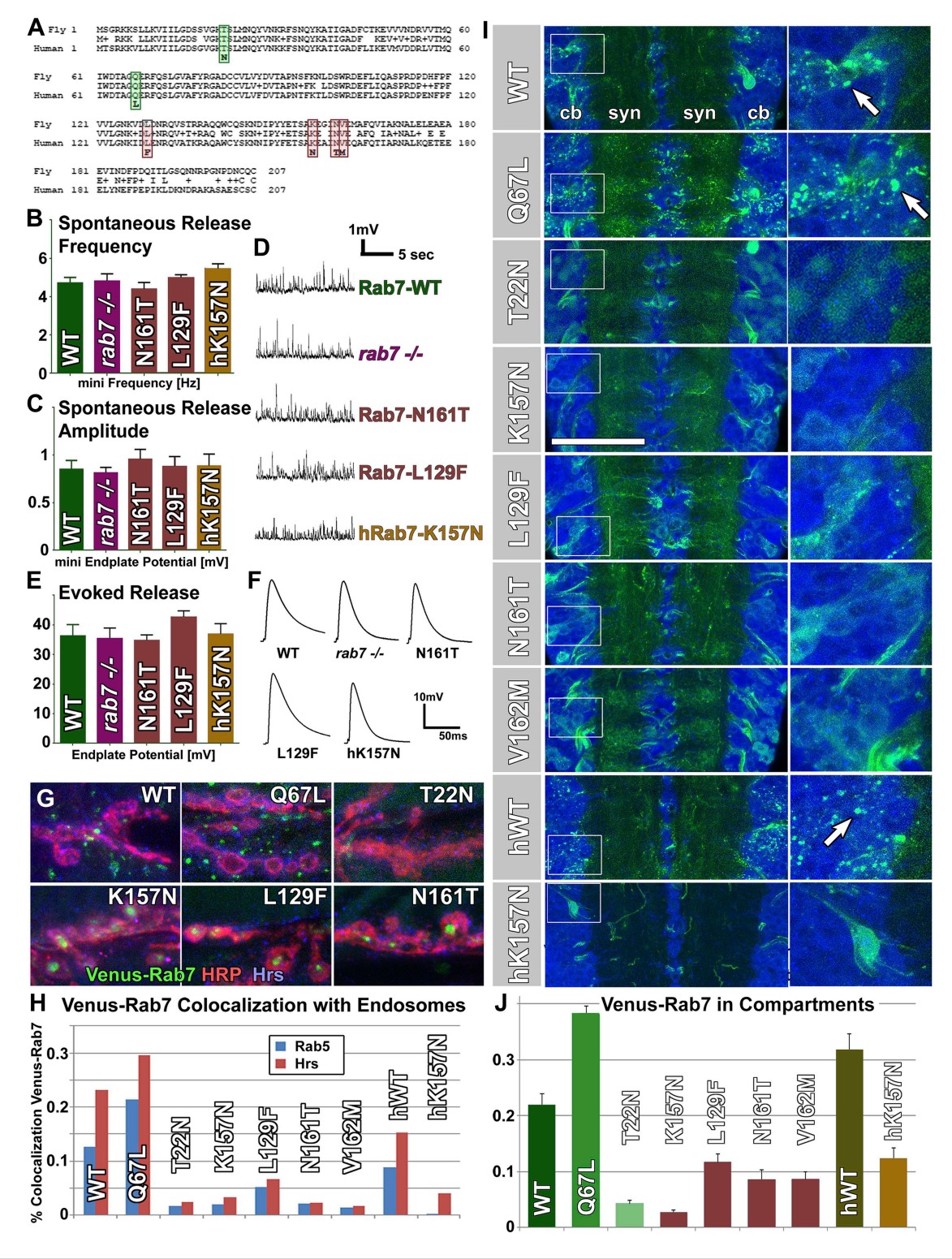

**Figure 2**. Overexpression of Venus-tagged Rab7 variants in motor neurons. (**A**) Protein alignment of fly and human Rab7 reveals 100% conservation of protein length and the precise locations of all CMT2B mutations (red) and the classically designated 'dominant negative' and 'constitutively active' mutations (green). (**B–F**) Electrophysiological recordings from the larval neuromuscular junction for ctrl, the *rab7* null mutant and overexpression of *rab7^N161T^*, *rab7^L129F^* and the human disease gene *hrab7^K157N^*. Spontaneous vesicle release (minis) exhibit normal frequency (**B**) and amplitude (**C**). Similarly, evoked neurotransmission is indistinguishable for all genotypes (**E** and **F**). (**G**) Synaptic boutons at the larval neuromuscular junction, immunolabeled for

*Figure 2. Continued on next page*

*Figure 2. Continued*

the presynaptic membrane (HRP, red), the endosomal marker Hrs (blue) and six different Venus-Rab7 proteins. (**H**) Colocalization quantification for all nine Venus-Rab7 proteins at the neuromuscular junction with the endosomal markers Rab5 and Hrs. (**I** and **J**) Analysis of the subcellular protein localizations of all Venus-tagged Rab7 variants (green) in the larval ventral ganglion, as previously performed for all Rab GTPases (*Chan et al., 2011*). Blue (DNA labeled with Toto-3) indicates areas of cell bodies (cb) and synapses (syn) in two center stripes. (**J**) Quantification of Venus fluorescence signal in 3D datasets inside clearly discernable compartments (arrows in **I**) as ratio of total fluorescence signal. See 'Materials and methods' for details. Scale bar in (**I**): 50 μm.

The following figure supplements are available for figure 2:

**Figure supplement 1**. Western blot analysis of total protein extract from fly eyes.

Rab7 variants. Co-labeling with Rab5 and Hrs reveals significant overlap with Venus-Rab7-WT that is further increased for Venus-Rab7-Q67L (*Figure 2H*). In contrast, both Rab5 and Hrs colocalization are almost completely lost for Venus-Rab7-T22N as well as all CMT2B proteins (*Figure 2H*). In particular, the synaptic accumulations of CMT2B variants exclude endosomal markers, suggesting that they are not functional. These findings are consistent with the absence of electrophysiological defects of these synapses.

To further analyze the function of each of the nine Venus-Rab7 proteins, we analyzed their recruitment to distinct compartments vs diffuse localization in the ventral ganglion where the motor neuron cell bodies reside. The GTP-locked Rab7-Q67L has an increased activity that is quantitatively reflected in its localization to late endosomal compartments, whereas the GDP-locked Rab7-T22N is considered inactive, as reflected by a complete failure to be recruited to endosomal compartments. Hence, compartment localization is a quantitative readout for GTP-dependent Rab7 activity (*Mukhopadhyay et al., 1997a*; *Zhang et al., 2007*; *Chan et al., 2011*). As shown in *Figure 2I–J*, the nine variants exhibit localization to distinct compartments (arrows in *Figure 2I*) and diffuse labeling in varying ratios. Quantification of the ratio of compartment fluorescence to total fluorescence in 3D datasets of individual cell bodies shows that 38% of Rab7-Q67L, compared to 21% of wild-type Rab7 and less than 4% of Rab7-T22N localize to distinct compartments (*Figure 2J*). These data are consistent with a plethora of previous studies on these widely studied Rab7 variants (*Mukhopadhyay et al., 1997a*, *1997b*; *Bucci et al., 2000*; *Zhang et al., 2007*). Analysis of the Venus-tagged CMT2B variants reveals mostly a diffuse cytoplasmic location, similar to Rab7-T22N, with 2–11% of the total fluorescent protein localized to distinct compartments. Hence, the reduced compartment localization suggests that the CMT2B proteins retain ~5–50% of the function of Rab7-WT. A similar difference is observed for the human proteins hRab7-WT and hRab7-K157N (*Figure 2J*). These data do not support the hypothesis that the disease variants mimic the gain-of-function of the Q67L mutations. Instead, they suggest that CMT2B variants exhibit reduced function more similar to the T22N mutant, but may retain some wild-type protein function that, based on quantitative analysis of localization to distinct compartments, ranges between 5–50% of Rab7-WT.

## Prolonged overexpression of GTP-bound, GDP-bound or CMT2B variants does not cause neuropathy-like phenotypes in sensory neurons in vivo

Our findings indicate that *rab7* is not acutely required for synaptic function in motor neurons or photoreceptor sensory neurons. However, loss of *rab7* leads to progressive, stimulation-dependent loss of synaptic function, which can be studied in photoreceptor neurons over prolonged time periods. We therefore asked whether overexpression of any of the nine Venus-Rab7 variants causes neuropathy-like phenotypes over time in vivo. Protein localization of the Venus-Rab7 variants in the photoreceptor synaptic layer in the brain revealed that all four CMT2B proteins accumulate under the exclusion of other endosomal markers, very similar to our findings in motor neurons (*Figure 3A–B*, compare *Figure 2G*). In contrast, both Venus-Rab7-WT and Venus-Rab7-Q67L exhibit significant colocalization with both Rab5 and Hrs at synapses (*Figure 3B*). We also analyzed cell bodies of central nervous system neurons in the same brains. Here, the constitutively active Rab7-Q67L exhibits a typical increase of endosomal compartments positive for Rab7-Q67L, Rab5 and Hrs (*Figure 3C–D*). In contrast, the CMT2B variants exhibit strongly reduced Rab5 and Hrs colocalization. These findings resemble the findings in motor neurons and suggest that CMT2B proteins are mostly dissociated from functional endosomes.

Next we asked whether the CMT2B proteins impair sensory neuronal function over longer periods of time or in a usage-dependent manner. To challenge the photoreceptor neurons, we exposed the

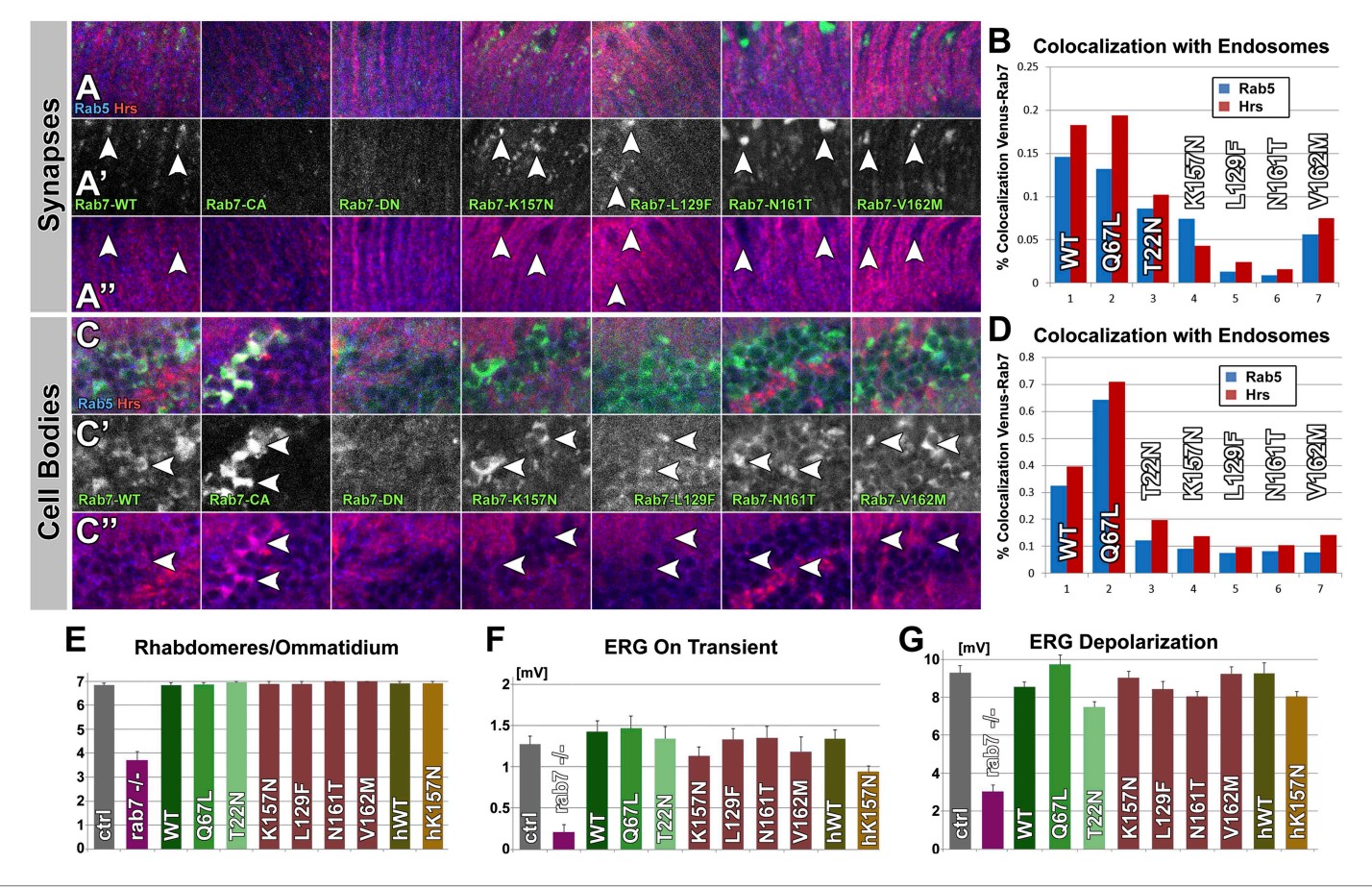

**Figure 3**. Overexpression of Venus-tagged Rab7 variants in photoreceptor sensory neurons. (A–D) Immuno-histochemical analyses of Venus-Rab7 protein localization and colocalization with the endosomal markers Rab5 (blue) and Hrs (red). (A) Longitudinal sections through the adult lamina, where photoreceptor neurons R1-R6 terminate. (C) Cell bodies of neurons in the medulla cortex. Green: Venus-Rab7 proteins, red: Hrs, blue: Rab5. (B and D) Quantification of Venus-Rab7 colocalization with Rab5 and Hrs for the indicated genotypes at synapses (B) and in cell bodies (D). (E and F) Quantification of morphological and functional analyses of Rab7 overexpression for all nine variants reveals a loss of endosomal colocalization for all CMT2B mutants. (genotype: UAS-*rab7-X*/+; *rab7*[Gal4-knock-in]/+) (E) The numbers of rhabdomeres per ommatidial cross section reveal that overexpression of none of the *rab7* mutants leads to morphological disruption similar to the mutant. (F and G) Overexpression of none of the mutant *rab7* variants causes defects in ERG depolarization or synaptic transmission ('on' transient). Control and overexpression experiments exhibit no statistically significant variance (ANOVA). Picture and ERG traces for all genotypes are shown in *Figure 3—figure supplement 1*.

The following figure supplements are available for figure 3:

**Figure supplement 1**. ERG recordings from overexpression experiments of the indicated *rab7* mutant variants at identical levels using *rab7*[Gal4-knock-in] in heterozygosity.

flies to 10 days of constant light stimulation. In stark contrast to the null mutant, neither overexpression of *rab7*[Q67L], *rab7*[T22N] nor any of the CMT2B variants in the *Drosophila* or human Rab7 protein causes any obvious morphological or functional defects (*Figure 3E–G*, *Figure 3—figure supplement 1*). In summary, our findings are not consistent with the gain-of-function hypothesis. None of the mutant Rab7 proteins exhibit obvious dominant negative characteristics at 2–3-fold overexpression in the presence of wild-type Rab7 in vivo.

## All CMT2B mutant proteins retain sufficient wild-type function to rescue the *rab7* null mutant

To measure the wild-type function or a putative dominant deleterious effects of each of the *rab7* variants we designed a quantitative population experiment. We established nine stocks that are both

heterozygous for the null mutant *rab7^{Gal4-knock-in}* allele and heterozygous for each of the nine UAS-*rab7* transgenes. Note that the *rab7^{Gal4-knock-in}* is homozygous lethal, but can become homozygous viable in the presence of one or two copies of the rescuing UAS-*rab7* transgenes (compare *Figure 1H*). Hence, four genotypes are possible in each of the nine stocks (*Figure 4A*): the heterozygotes with overexpression of one (light blue) or two (dark blue) copies of a UAS-*rab7* variant and the null mutant homozygotes, rescued by one (light green) or two copies (dark green) of a UAS-*rab7* variant.

We incubated the nine stocks for four generations and counted the relative populations. The ratio of genotypes that emerge after four generations is indicative of the fitness of each genotype relative to the others in the population. For instance, if overexpression of any of the nine *rab7* variants has deleterious effects, two copies of the transgene will be less common than one. In the case of wild-type Rab7, three quarters of the population had two copies of the Rab7-WT transgene, indicating little to no toxicity of the overexpressed Rab7-WT. Furthermore, 12% of the flies in the population were rescued homozygous *rab7* null mutants (light and dark green in *Figure 4A*). We note that this number reflects a remarkably healthy rescue, because a significant part of the population lives without the wild-type *rab7* chromosome, but instead only due to Rab7-Gal4-driven UAS-*rab7^{WT}* in direct competition with wild-type flies over four generations. Indeed, the ratio of rescued to wild-type progeny prior to the competition experiment obeyed Mendelian ratios and homozygous *rab7* null mutant flies rescued with Rab7-WT expression (UAS-*rab7^{WT}*/UAS-*rab7^{WT}*; *rab7^{Gal4-knock-in}*/*rab7^{Gal4-knock-in}*) appear indistinguishable from wild type.

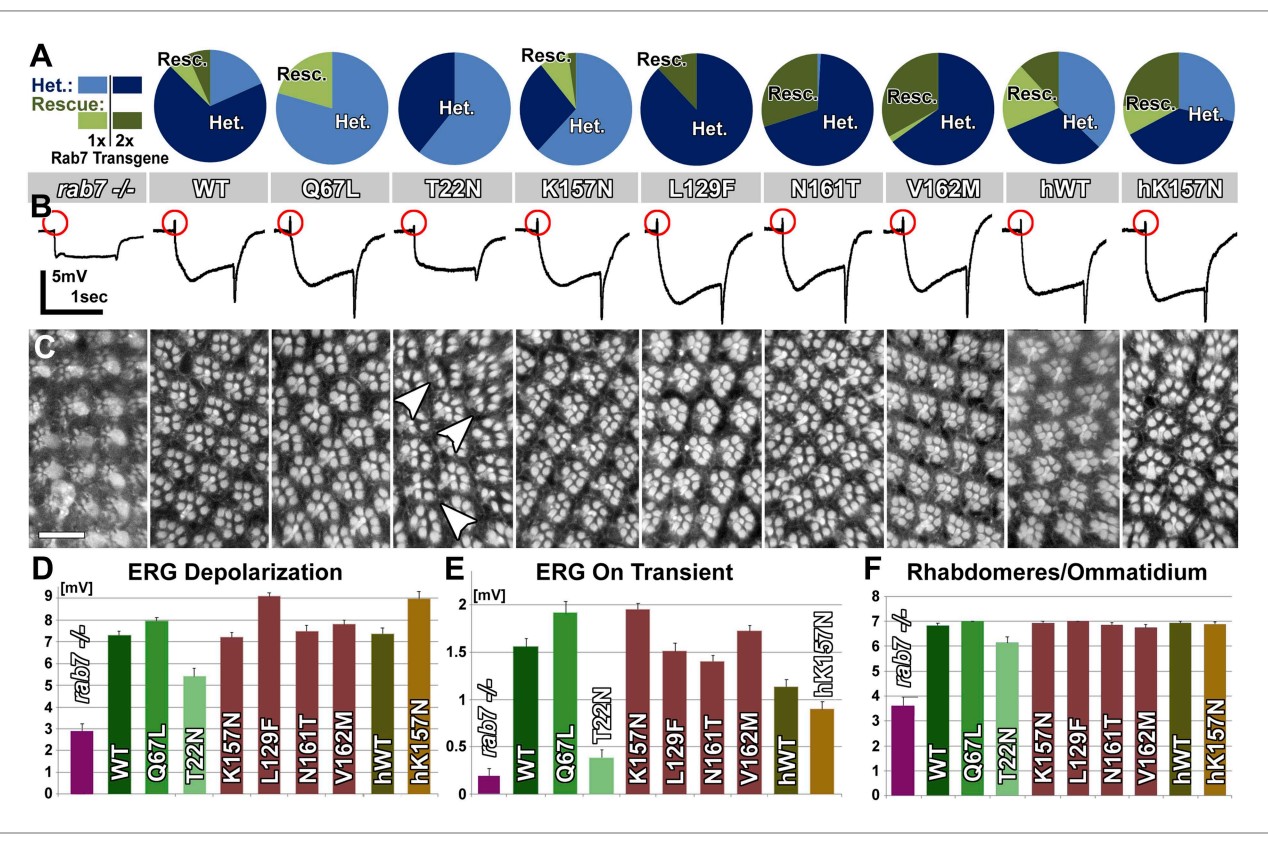

Figure 4. Rescue experiments using *rab7^{Gal4-knock-in}* driven UAS-venus-*rab7* variants. (A) A population experiment over four generations reveals the fittest and unhealthy genotypes. Light Blue shows the fraction of flies heterozygous for the null mutant and expressing 1 copy (2–3-fold overexpression) for each of the nine transgenes (genotype: UAS-*rab7-X*/+; *rab7^{Gal4-knock-in}*/+). Dark Blue shows the fraction of heterozygous *rab7* flies expressing two copies of the respective transgenes (genotype: UAS-*rab7-X*/UAS-*rab7-X*; *rab7^{Gal4-knock-in}*/+). Light green shows the fraction of homozygous null mutant flies rescued through expression of one copy (2–3-fold overexpression) of the respective transgene (genotype: UAS-*rab7-X*/+; *rab7^{Gal4-knock-in}*/*rab7^{Gal4-knock-in}*). Dark green shows the fraction of null mutant flies rescued by two copies (4–6-fold) overexpression of one of the transgenes (genotype: UAS-*rab7-X*/UAS-*rab7-X*; *rab7^{Gal4-knock-in}*/*rab7^{Gal4-knock-in}*). (B–F) Rescue experiments of synaptic and neuronal degeneration using each of the nine *rab7* transgenes in null mutant photoreceptors after 10 days of constant light stimulation. (B) Representative ERG traces with measured components in (D) and (E). (C) Representative eye cross sections showing the array and number of rhabdomeres per ommatidium, and corresponding counts in (F). Variance for measurements in (D–F) is significantly different for *rab7* null mutant and *rab7^{T22N}* rescue (ANOVA). Scale bar in (C): 10 μm.

In the case of Rab7-T22N 40% of the population had two copies of this widely used 'dominant negative' transgene, but no rescued homozygous *rab7* null mutants were observed. Hence, Rab7-T22N exhibits little or no toxicity but is also not sufficiently functional to efficiently rescue the null mutant. Interestingly, close investigation of the fly stock revealed that the *rab7$^{T22N}$* mutant actually yielded rare adult escapers that died shortly after emergence. In contrast, Rab7-Q67L never occurred in two copies, indicating that high levels of this constitutively active mutant are associated with reduced fitness (*Figure 4A*). In contrast, two of the four CMT2B variants lead to populations consisting entirely of individuals expressing two copies of these disease-associated proteins. Remarkably, almost no rescue of the null mutant with a single copy of these CMT2B transgenes was observed, but between 12–35% of individuals were rescued by two copies of the CMT2B mutant transgenes. These findings suggest that higher levels of at least three of the four CMT2B mutant proteins increase the fitness and are indeed required to compensate for the loss of wild-type *rab7*. In summary, all CMT2B disease mutants, including the *Drosophila* and human Rab7-K157N, exhibited significant rescue (*Figure 4A*). Indeed, all UAS-*rab7* variants except Rab7-T22N rescued *rab7* lethality and yielded adult flies with no obvious defects. These findings reveal that the CMT2B proteins do not reduce fitness or neuronal health, but retain sufficient wild-type function to compensate for loss of *rab7* if expressed at 2–6-fold endogenous levels.

To measure the ability of all *rab7* variants to rescue the progressive and usage-dependent synaptic and neuronal degeneration observed in the null mutant we measured ERGs in photoreceptors after 10 days constant light exposure. As shown in *Figure 4B–F*, all *rab7* variants rescued the null mutant pheno-types. Only *rab7$^{T22N}$* exhibited an only partial ability to rescue the response amplitudes (*Figure 4B,D*), synaptic function (*Figure 4B,E*) and rhabdomere morphology (*Figure 4C,F*). However, the observation that *rab7$^{T22N}$* exhibits some rescue of the null mutant further indicates that it retains some wild-type function and does not obviously act as a genetically dominant negative mutant at these levels of overex-pression in Drosophila. While both human Rab7 proteins exhibit a reduced ability to rescue synaptic functions under these stress conditions, *hrab7$^{K157N}$* does not significantly differ from *hrab7$^{WT}$*. The ability of all CMT2B mutant proteins to rescue the null mutant phenotype indicates that replacement of wild-type *rab7* with mild overexpression in the correct spatiotemporal pattern is sufficient to com-pensate for the partial loss of function of the mutant variants. In summary, we conclude that all CMT2B mutants retain significant levels of wild-type function.

## Live imaging reveals that CMT2B proteins are inefficiently recruited to endosomes, but do not dominantly impair endosomal maturation

What causes the reduced wild-type function of the CMT2B mutant proteins? To measure the function of each of the Rab7 variants, we devised a live imaging method to assay the well-characterized role of Rab7 in the maturation of late endosomal compartments. We developed a culture system for high-resolution live imaging of endosomal dynamics in photoreceptor neurons in intact eye-brain complexes and made use of the lysosomal marker spinster/benchwarmer (spin) (*Sweeney and Davis, 2002*; *Dermaut et al., 2005*). Co-expression of the Venus-tagged Rab7 proteins with spin-RFP reveals the dynamics of endolysosomal conversion both in cell bodies and at synapses (*Videos 1–10*). In cell bodies, Venus-Rab7-WT marks on average 15 clearly distinguishable, circular endosomal compart-ments of 0.5 µm diameter or more per 500 µm² tissue area. In contrast, Rab7-T22N marks no individual compartments and expression of the constitutively active Venus-Rab7-Q67L leads to a doubling of compartment numbers, many of which are significantly larger than those observed for wild type (*Figure 5A–B*). As seen before, all four CMT2B mutant proteins exhibit mostly diffuse labeling similar to Rab7-T22N; however, in contrast to Rab7-T22N and Rab7-K157N, the other three CMT2B variants also reveal clear circular labeling of distinct compartments of 0.5 µm diameter or above (*Figure 5C*), albeit at significantly reduced numbers (*Figure 5G,H*). 3D time lapse imaging of the dynamics of these compartments reveals conversion of 8–9% of these compartments into spin-RFP positive compartments within a 5 min imaging interval. Conversion of wild type and CMT2B Venus-Rab7 marked compartments typically occurs quickly and within the 30 s between image acquisition time points (*Figure 5D–F*). Remarkably, the conversion rate of 8–9% per 500 µm² tissue area per 5 min is identical between all genotypes (*Figure 5I*). Based on these measurements, the CMT2B proteins exhibit 5–50% of the wild type levels of recruitment to endosomal compartments, but no defects in endosomal conversion.

Taken together, these live imaging experiments reveal that the CMT2B proteins are poorly functioning proteins; their inefficient recruitment to endosomes provides a mechanistic basis for their partial loss-of-function. However, the CMT2B proteins do not dominantly affect the conversion rate of endosomal

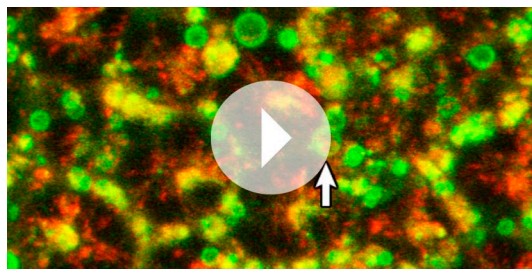

**Video 1**. Venus-Rab7-Q67L and spin-RFP in the eye.

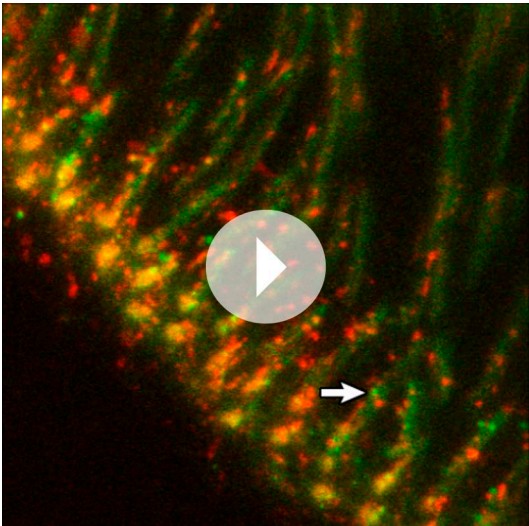

**Video 2**. Venus-Rab7-Q67L and spin-RFP in photoreceptor terminals.

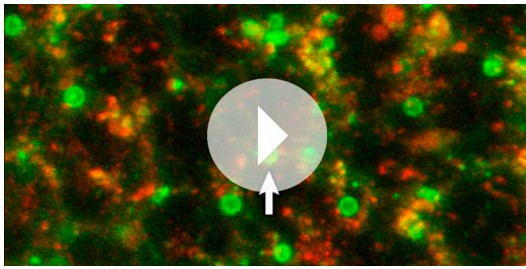

**Video 3**. Venus-Rab7-WT and spin-RFP in the eye.

compartments they are recruited to, corroborating the findings that the CMT2B proteins do not dominantly impair the function of wild-type Rab7.

## Partial loss of *rab7* function leads to progressive, stimulation-dependent degeneration

Based on our findings we propose a partial loss-of-function model for CMT2B in which partial or complete loss of one copy of *rab7* causes degeneration. This model predicts that sensory neurons in vivo are sensitive to the precise dosage of *rab7*-dependent endolysosomal degradation. We therefore analyzed viable heterozygous flies for increased sensitivity to stimulation using 10-days of light exposure in white-eyed flies. As shown in *Figure 6A,B*, constant stimulation leads to almost complete loss of synaptic function in heterozygous *rab7* mutants similar to the null mutant. This defect is fully rescued by the absence of stimulation and does not significantly affect the photoreceptor response amplitude (*Figure 6A,B*). Furthermore, *rab7* heterozygous photoreceptors after stimulation exhibit a morphological haploinsufficiency phenotype with partial loss of rhabdomere structures (*Figure 6C–F*). These findings indicate that partial loss of *rab7* affects the function of these sensory neurons in a stimulation-dependent manner without any other obvious effects on development, viability or other tissues in *Drosophila*.

## CMT2B protein levels at and below endogenous Rab7 levels are sufficient to rescue the *rab7* haploinsufficiency in *Drosophila* photoreceptor neurons

Our findings indicate that neurons are sensitive to the precise levels of Rab7. If the CMT2B mutations are hypomorphic alleles and encode proteins with 5–50% function, as our data indicates, then the cause of the adult-onset loss of synaptic function in patients may be due to this partial loss of function. On the other hand, numerous overexpression experiments revealed clear dominant effects of the CMT2B proteins in cell culture (*Spinosa et al., 2008*; *Cogli et al., 2010*; *McCray et al., 2010*; *Basuray et al., 2013*; *Cogli et al., 2013*; *Zhang et al., 2013*). To determine the levels and range of CMT2B protein levels that are sufficient to rescue partial loss of *rab7* or induce dominant effects, we generated heterozygous *rab7* flies (one copy of the wild-type chromosome) and a second copy at half, identical and >10-fold levels of the endogenous copy of *rab7*. To generate these flies, we introduced the temperature-sensitive Gal4 suppressor Gal80ts (*McGuire et al., 2003*) and determined temperature conditions to match the CMT2B overexpression levels to endogenous Rab7 levels and further reduce the CMT2B expression 50% below endogenous levels. As shown in *Figure 7A*, an antibody against *Drosophila* Rab7 (*Chinchore et al., 2009*) recognizes both endogenous Rab7 as well as our Venus-tagged transgenes. More than 10-fold

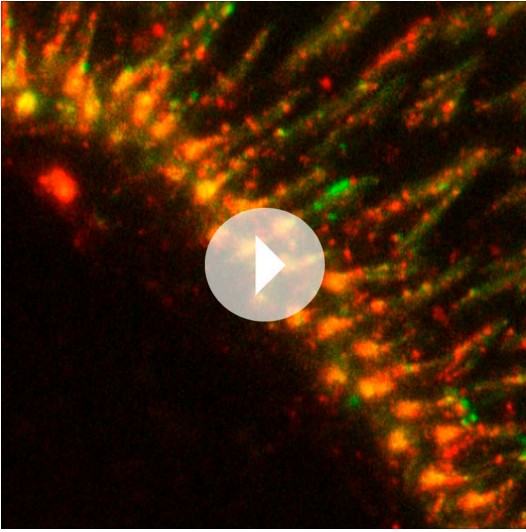

**Video 4**. Venus-Rab7-WT and spin-RFP in photoreceptor terminals.

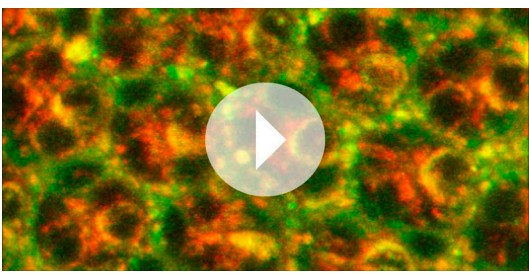

**Video 5**. Venus-Rab7-V162M and spin-RFP in the eye.

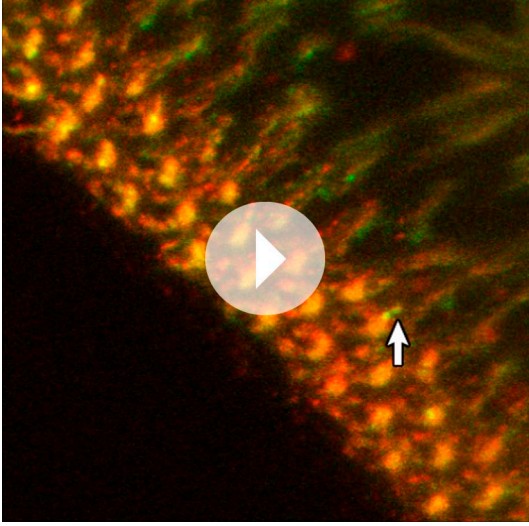

**Video 6**. Venus-Rab7-V162M and spin-RFP in photoreceptor terminals.

overexpression of the CMT2B variants was achieved in the absence of Gal80ts at 25°C. Remarkably, all transgenes over this large range rescued the haploinsufficiency phenotype after a 5-day light stimulation protocol in the ERG depolarization (*Figure 7B*), synaptic function (*Figure 7C*) and Rhabdomere structure of the eye (*Figure 7D–J*). These findings quantitatively support two key findings reported in this study: First, Rab7 CMT2B proteins at 50% less than endogenous heterozygous levels (corresponding to 25% of total Rab7 function) retain sufficient wild-type function to rescue the *rab7* phenotype. Second, Rab7 CMT2B proteins at >10-fold overexpression rescue, but exhibit no toxic or other obvious gain-of-function effects. Together with our quantitative colocalization and live imaging experiments that show quantitatively reduced wild-type function, these data corroborate our conclusion that CMT2B proteins represent partial loss-of-function alleles with no toxic gain-of-function in *Drosophila*.

## Discussion

In this study we present a comprehensive analysis of *rab7* loss- and gain-of-function at defined expression levels in motor neurons and sensory neurons in vivo. Our results show that loss of *rab7* function causes progressive and usage-dependent loss of synaptic function and neurodegeneration without affecting neuronal development or other tissues. This loss-of-function phenotype is dosage-dependent. In contrast, overexpression of the CMT2B mutant variants of *rab7* causes no neuropathy-like phenotypes, even at high levels. We conclude that the four CMT2B mutants are hypomorphic alleles of *rab7* that retain 5–50% of wild-type function (*Figure 8A,B*). We therefore propose that CMT2B is a dominant neuropathy due to partial loss-of-function. This reinterpretation of CMT2B has the significant consequence that a potential therapy may involve increasing the mutant protein function, which is opposite to the current proposal to reduce the mutant protein function.

### Evidence for the partial loss-of-function of CMT2B alleles

Despite the identification of the CMT2B mutations as hypomorphic alleles, our findings are remarkably consistent with the majority of previous biochemical findings. For example, mammalian heterologous expression studies revealed that the GTP: GDP ratio is altered, which could suggest a dominant or constitutively active mutant. However, the same study revealed that, in contrast to the constitutively active variant, the total amount of both GTP

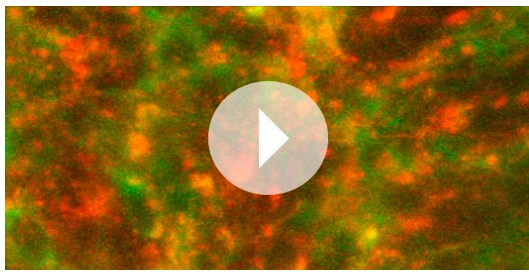

**Video 7**. Venus-Rab7-K157N and spin-RFP in the eye.

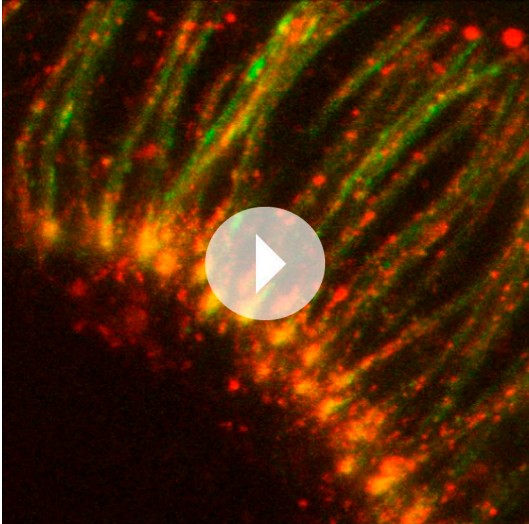

**Video 8**. Venus-Rab7-K157N and spin-RFP in photoreceptor terminals.

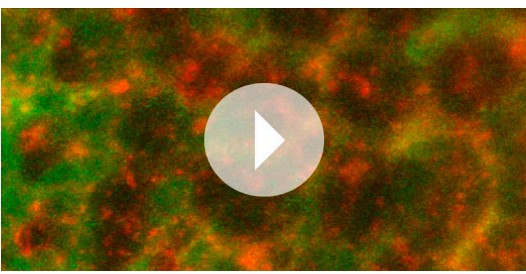

**Video 9**. Venus-Rab7-T22N and spin-RFP in the eye.

and GDP bound Rab7 is dramatically decreased for the CMT2B mutant proteins (*De Luca et al., 2008*; *Spinosa et al., 2008*). Furthermore, our findings are largely consistent with the thorough biochemical characterization of the CMT2B Rab7 proteins by *McCray et al. (2010)*. Specifically, McCray et al. found that the CMT2B proteins have no GTPase activity defect, but augmented protein activity, based on increased guanine nucleotide dissociation, hydrolysis-independent inactivation, quantitative changes on effector interactions and decreased membrane cycling. However, McCray et al. also showed that the CMT2B proteins retain sufficient wild-type function to rescue reduced *rab7* activity in HeLa cells without obvious toxic gain-of-function effects. All these findings are consistent with partial loss of function alleles. The key difference to the present study lies in the interpretation that misregulation of the disease proteins' Rab7 activity may have a dominant effect, as opposed to our simpler interpretation as partial loss-of-function.

## Limitations of the *Drosophila* model

Our findings do not provide an obvious explanation for recent reports on specific dominant effects of CMT2B protein overexpression in cell culture (*Spinosa et al., 2008*; *Cogli et al., 2010*; *McCray et al., 2010*; *Basuray et al., 2013*; *Cogli et al., 2013*; *Zhang et al., 2013*). These dominant phenotypes could be a result of overexpression in a heterologous cell line, or a *bona fide* property of the CMT2B proteins. In some cases, high levels of overexpression might at least partially explain the effects. Nonetheless, our data do not rule out the possibility of a dominant interaction with protein complexes or protein functions that are specific to vertebrate cells. However, we note that none of the studies to date has reported progressive neuronal or synaptic deterioration as a consequence of CMT2B protein overexpression in motor or sensory neurons (*Cogli et al., 2013*; *Zhang et al., 2013*). While we show here that such putative vertebrate-specific properties are not necessary to cause the neuropathy in a *Drosophila* model, they may contribute to specific neuronal changes over time in the different heterologous cell culture systems as well as in patients.

A second limitation of the *Drosophila* model is revealed by our effort to precisely mimic the human patient genotypes that lead to identical levels of expression of one wild type copy of Rab7 and one CMT2B mutant variant. In patients, this genotype leads to slow motor- and sensory neuron degeneration with an onset between 12 to more than 40 years. In *Drosophila*, heterozygotes with an additional copy of a CMT2B chromosome are healthy and photoreceptor neurons are not sensitized to stimulation-dependent degeneration within the limits of our functional and morphological assays. The observation that this genotype is sufficient for normal human development and function for decades offers a possible explanation for this limitation of the fly model. Indeed, slow, adult-onset degeneration over

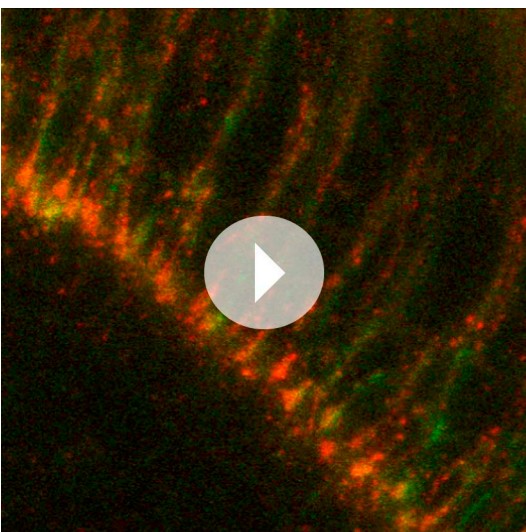

**Video 10**. Venus-Rab7-T22N and spin-RFP in photoreceptor terminals.

a period of years is not easily mimicked in any model organism. It will be interesting to see if a *rab7* heterozygous mouse model exhibits stimulation-dependent degeneration of motor- or sensory neurons over time, or if even the lifespan of a mouse is not sufficient to model this aspect of the human neuropathy.

## The partial loss-of-function model

We propose that CMT2B in patients reflects a *rab7* dosage-dependence, as described here in a fly model. Since no patients have been reported with a null mutant allele of *rab7*, we speculate that complete loss of one copy of *rab7* in humans may cause lethality (**Figure 8C**). Adding a second *rab7* allele with 5–50% function may be sufficient to retain normal function in most cells but slowly cause defects over a period of years only in the cells most sensitive to *rab7*-dependent endolysosomal degradation (**Figure 8C**). Our model predicts that only mutations that reduce *rab7* function within a certain range will lead to a neuropathy, explaining the rarity of these mutations and the variability of disease onset. We note that the CMT2B mutant with the least function in the fly (**Figure 8A**), *rab7K157N*, is a sporadic new mutation in a patient with the earliest reported childhood onset (12 years) of the four mutant variants (**Meggouh et al., 2006**).

Mild reductions in endolysosomal degradative capacity may be caused by numerous genetic polymorphisms as well as an increased degradative load in various degenerative disorders characterized by intracellular accumulations. Such accumulations and subsequent endomembrane degradative responses are hallmarks of most neurodegenerative disorders. An elevated neuronal demand for endolysosomal degradation is further highlighted by the recent discovery of a neuron-specific branch of the endolysosomal system (**Williamson et al., 2010**; **Haberman et al., 2012**; **Wang and Hiesinger, 2012**). In the case of CMT2B our findings suggest an increase of endolysosomal function as a therapeutic approach, which is opposite to the current proposal to reduce the function of the mutant Rab7 proteins. Neuronal sensitivity to *rab7*-dependent degradation may be a common factor contributing to neuronal pathology in numerous disorders with reduced degradation or increased degradative burden. Reduced endolysosomal capacity may thus contribute to pathology and increased endolysosomal function may represent a more general therapeutic opportunity.

## Materials and methods

### Generation of the *rab7* knock-out

The generation of *rab7* targeting vector by recombineering was performed as previously described (**Chan et al., 2011**, **2012**) with some modifications. We generated a new genomic fragment containing the *rab7* locus with asymmetric flanking homology arms of 10 kb at 5' and 5 kb at 3'. The final targeting vector was injected into PBac{y[+]-attP-3B}VK00033 landing site by Rainbow Transgenic Services (CA, USA) using PhiC31-mediated integration.

The knock-out screen was performed according to published protocols (**Chan et al., 2011**, **2012**). The mobilization and reintegration by ends-out homologous recombination was initiated by heat shock. We screened approximately 80,000 F2 progeny for separation of the targeting cassette from the original landing site and identified 61 reintegrations in the genome. 58 of the 61 genomic integration events occurred on the third chromosome, the correct *rab7* bearing chromosome. 12 lethal lines die at late pupal stage. To characterize the potential knock-outs molecularly, the genomic DNA of homozygous larvae was extracted and examined by PCR. The 46 viable lines showed no correct replacement. Of the 12 lethal lines, nine failed to complement two independent deficiencies uncovering the *rab7* locus (Df(3R)ED10893, 95C8;95E1, 3R:19713027;19930781 and Df(3R)Exel6196

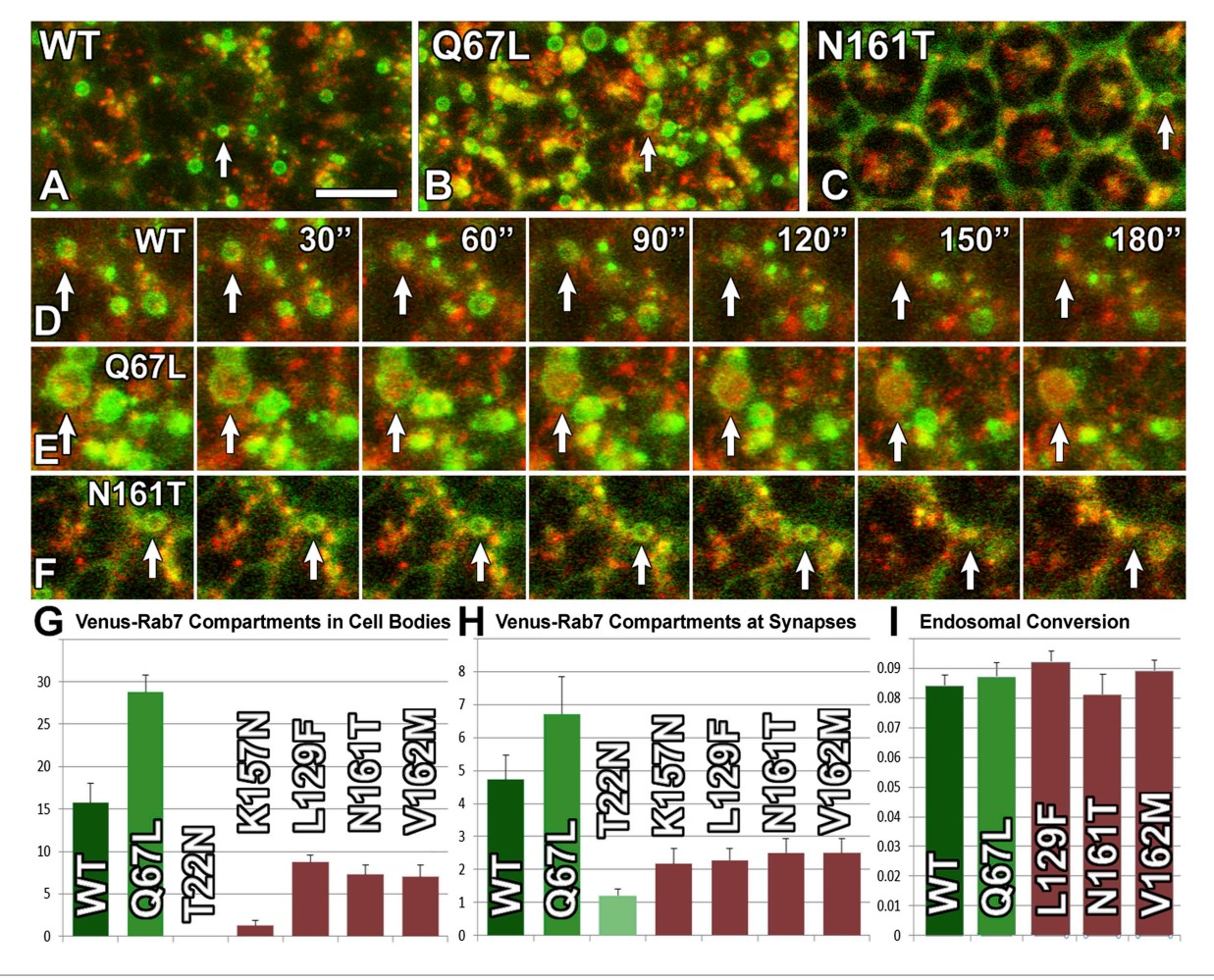

**Figure 5**. Live imaging reveals that CMT2B Rab7 proteins exhibit defective endosomal recruitment but do not affect endosomal maturation. (**A**–**F**) Snapshots from live imaging datasets for three genotypes with a time lapse of 30 s (**A** and **D**) Venus-Rab7-WT and spin-RFP; (**B** and **E**) Venus-Rab7-Q67L and spin-RFP; (**C** and **F**) Venus-Rab7-N161T and spin-RFP. Arrows mark individual Rab7-positive compartments >=0.5 µm diameter. The same compartments marked in (**A**–**C**) are followed over time in (**D**–**F**). (**G**) Quantification of compartments as in (**A**–**F**) for all genotypes per 500 µm² area. (**H**) Quantification of Venus-Rab7 compartments (individual distinguishable green punctae) per photoreceptor axon terminal in the brain for the indicated genotypes. (**I**) Quantification of the fraction of compartments that underwent green-to-red conversion over a 5 min period in same data as (**G**). Note that Rab7-T22N and Rab7-K157N did not mark sufficient compartments for this analysis. See also: *Videos 1–10*. Scale bar in (**A**): 10 µm.

95C12;95D8, 3R:19747854-19747855;19857149). Two knock-out lines were verified by rescue experiments as described in the manuscript.

## Generation of nine UAS-venus-*rab7* constructs and transgenic lines

Mutagenesis of *rab7* wild-type cDNA to generate the four CMT2B mutations (L129F, K157N, N161T, and V162M) was conducted by PCR SOEing. These four mutant cDNAs, the wild-type cDNA as well as the previously generated T22N and Q67L variants (*Zhang et al., 2007*) were cloned into the pTVW vector (The *Drosophila* Gateway Vector Collection, Carnegie Institution) to generate N-terminal Venus fusions. For human *hrab7A^wt* and *hrab7A^K157N*, Venus and hRab7 were PCR amplified separately and then fused using SOEing PCR to generate the N-terminal Venus fusions. All nine *venus-rab7* variants were subcloned using PCR-generated Not1/Xho1 sites and ligated into the pUASt-attB (*Bischof et al., 2007*) for subsequent PhiC31-integrase mediated insertion in the PBac{y[+]-attP-3B}VK00002 landing site by Rainbow Transgenic Services (CA, USA). The following nine stocks were generated:

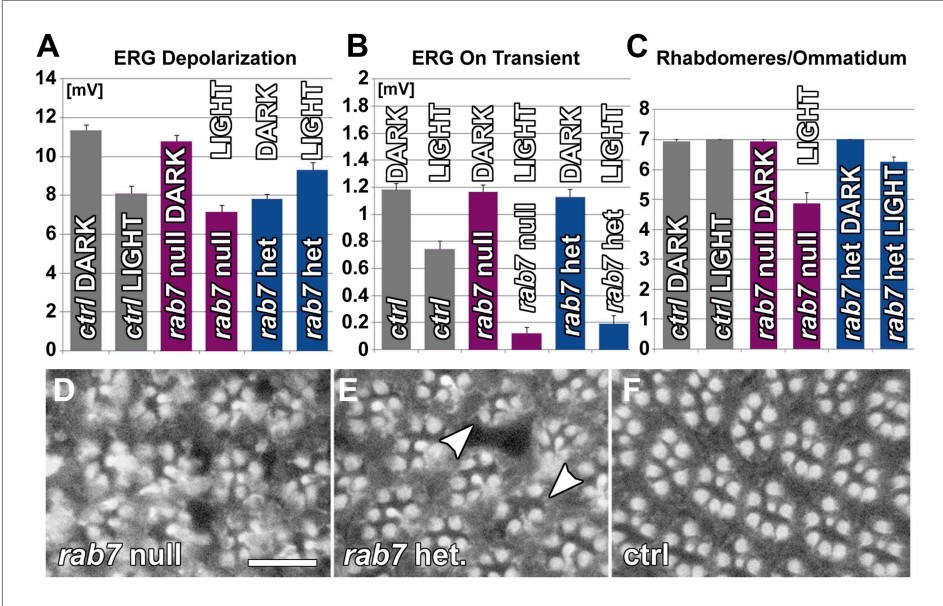

**Figure 6**. Photoreceptor neurons are haploinsufficient for *rab7* function as revealed by functional and morphological measurements 10 days after constant light stimulation. (**A** and **B**). Quantification of ERG depolarization and synaptic transmission ('on' transients) in null mutants and heterozygotes for *rab7*. (**C–F**) Ommatidial rhabdomere composition from eye cross sections exposed to various experimental conditions. Scale bar in (**D**) for (**D–F**): 10 μm.

- Wild type: yw; UAS-Venus-Rab7-WT/CyO; FRT82B *rab7*<sup>Gal4-knock-in</sup>/TM3, Sb
- GTP-bound 'Constitutively active': yw; UAS-Venus-Rab7-Q67L/CyO; FRT82B *rab7*<sup>Gal4-knock-in</sup>/TM3, Sb
- GDP-bound 'Dominant Negative': yw; UAS-Venus-Rab7-T22N/CyO; FRT82B *rab7*<sup>Gal4-knock-in</sup>/TM3, Sb
- CMT2B mutant: yw; UAS-Venus-Rab7-K157N/CyO; FRT82B *rab7*<sup>Gal4-knock-in</sup>/TM3, Sb
- CMT2B mutant: yw; UAS-Venus-Rab7-L129F/CyO; FRT82B *rab7*<sup>Gal4-knock-in</sup>/TM3, Sb
- CMT2B mutant: yw; UAS-Venus-Rab7-N161T/CyO; FRT82B *rab7*<sup>Gal4-knock-in</sup>/TM3, Sb
- CMT2B mutant: yw; UAS-Venus-Rab7-V162M/CyO; FRT82B *rab7*<sup>Gal4-knock-in</sup>/TM3, Sb
- Human rab7A wild type: yw; UAS-Venus-hRab7A-WT/CyO; FRT82B *rab7*<sup>Gal4-knock-in</sup>/TM3, Sb
- Human rab7A CMT2B mutant: yw; UAS-Venus-hRab7A-K157N/CyO; FRT82B *rab7*<sup>Gal4-knock-in</sup>/TM3, Sb

## Primers for mutagenesis SOEing PCRs

rab7-Fwd-ATG: ATG TCC GGA CGT AAG AAA TC; rab7-Rev-Stp: TTA GCA CTG ACA GTT GTC AG; L129F-Fwd: ATA AGG TGG ATT TCG ACA AC; L129F-Rev: TGG CGG TTG TCG AAA TCC AC; K157N-Fwd: AAA CGT CCG CCA ACG AGG GC; K157N-Rev: TTG ATG CCC TCG TTG GCG GA; N161T-Fwd: AGG AGG GCA TCA CCG TGG AG
N161T-Rev: GCC ATC TCC ACG GTG ATG CC; V162M-Fwd: AGG GCA TCA ACA TGG AGA TG; V162M-Rev: AAC GCC ATC TCC ATG TTG AT.

## Primers to amplify venus-*rab7*-wt/ca/dn/CMT2B mutants for cloning into pUASt-attB vector

V-rab7-fwd-Not1: ATA AGA ATG CGG CCG CAC CAT GGT GAG CAA GGG CGA G; V-rab7-rev-Xho1: CCG CTC GAG TTA GCA CTG ACA GTT GTC.

## Primers for human hrab7A SOEing PCRs

V-rab7-fwd-Not1: ATA AGA ATG CGG CCG CAC CAT GGT GAG CAA GGG CGA G; venus-hrab7 Rev: TAGAGGTCATCCGGTGCTTGTACAGCTCGTCCATG; venus-hrab7 Fwd: GCTGTACAAGCA CCGGATGACCTCTAGGAAGAAAG; hrab7-Rev-XhoI: CCGCTCGAGTCAGCAACTGCAGCTTTCTG.

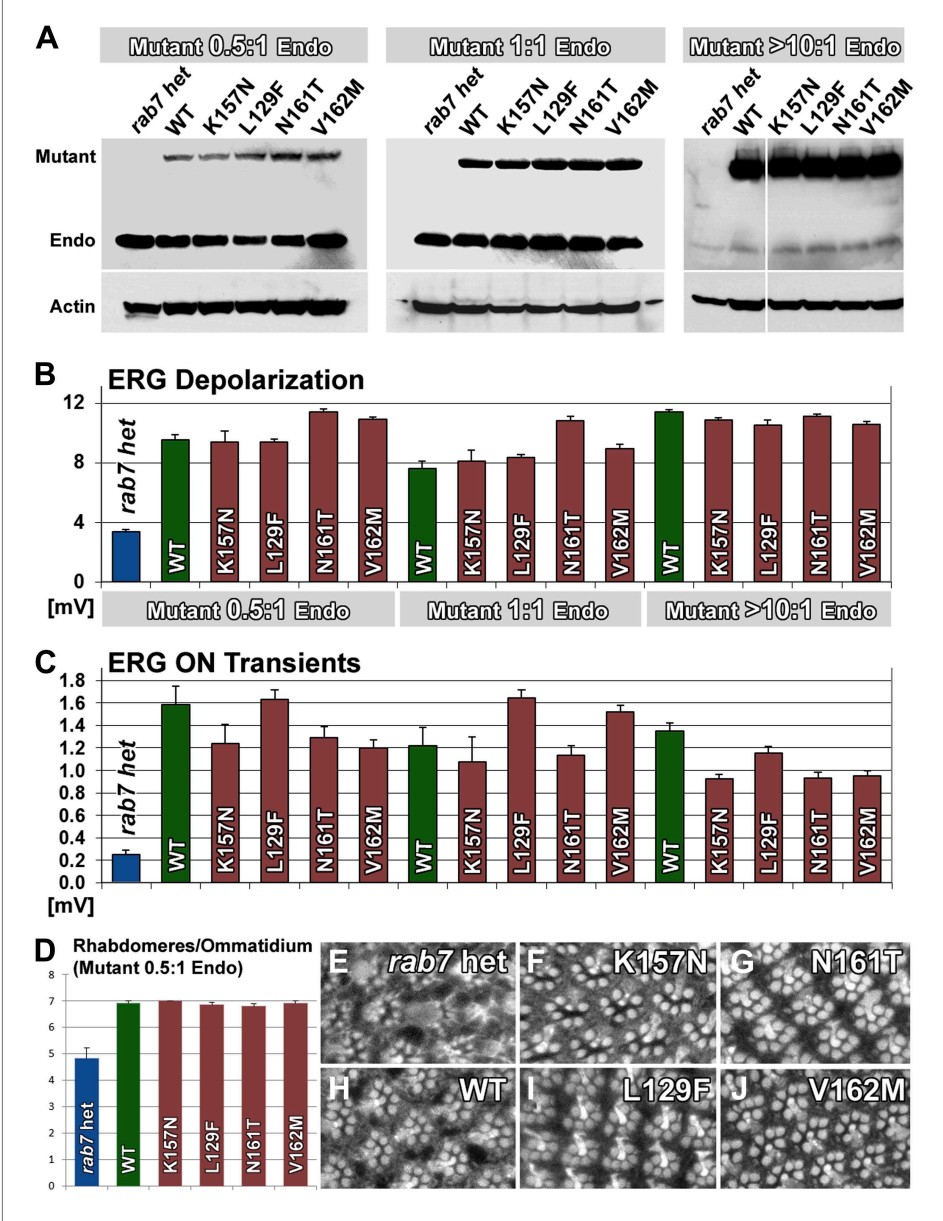

**Figure 7**. A wide range of CMT2B expression levels rescues partial loss of *rab7* function without dominant toxic effects. (**A**) Western blots for Rab7 of adult fly eyes expressing different levels of the Venus-tagged mutant transgenes. (**B** and **C**) ERG depolarization and On transient measurements after a 5-day light stimulation protocol. WT and CMT2B mutant values are not statistically significantly different in an ANOVA test. (**D**) Morphological analyses of rhabdomere structure in fly eyes with a ratio of 0.5:1 expression of the mutant transgenes vs endogenous heterozygous protein amounts. (**E**–**J**) Representative images of the rhabdomer structure for the indicated genotypes.

## Generation of the *rab7 Gal4 knock-in* targeting vector and knock-out screen

The primers used for amplifying the 500 bp homology arms, left arm (LA) and right arm (RA):

> LA Fwd ACAAGTTTGTACAAAAAAGCAGGCTTACAGTGTAGAAAGCAGCAA; LA Rev GGCAACGG ATCCTCACGTTGGTTTCGGAACAC; RA Fwd ACGTGAGGATCCGTTGCCACCGCTTCCTGCAT; RA Rev ACCACTTTGTACAAGAAAGCTGGGTACACCGTCACTCACTAGACC.

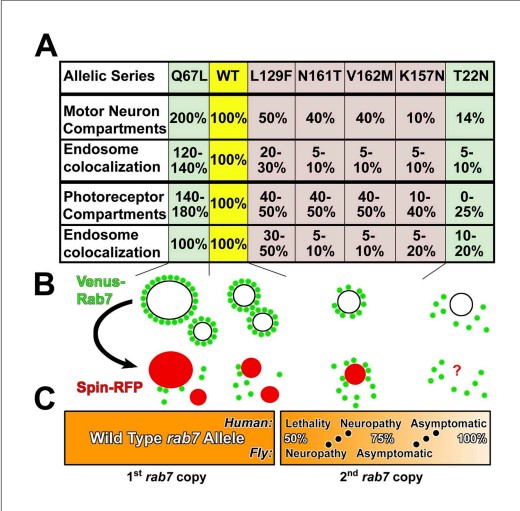

**Figure 8**. Summary of *rab7* allele function and partial loss-of-function model. (**A**) The seven *rab7* alleles analyzed here form an allelic series based on endosome recruitment and Rab5/Hrs colocalization in motor neurons and photoreceptor sensory neurons. First row summarizes data in *Figure 2J*; second row summarize data in *Figure 2H*; third row summarizes data in *Figure 5G,H* and the fourth row summarizes data in *Figure 3B,D*. Note that only the constitutively active Rab7-Q67L exhibits clearly reduced fitness at higher expression levels, but none of the CMT2B mutants (*Figure 4A*). (**B**) The allelic series is characterized by a gradual loss of the ability of Rab7 to be recruited to endosomal compartments. In contrast, conversion of Rab7-positive compartments is normal. (**C**) Partial loss-of-function model. A few day-old flies exhibit neuropathy-like phenotypes at 50% *rab7* function, which can be rescued by mildly increasing *rab7* function by expressing partial loss-of-function alleles. Humans exhibit neuropathy symptoms only after ≥12 years at levels that are, based on our analysis of the CMT2B alleles plus one wild type copy, between 60% and 90% of total *rab7* function.

The replacement of the *rab7* open reading frame with the Gal4knock-in cassette was performed by adding homology region to the Gal4knock-in cassette using the following primers:

5'-AAGAAACCATCACACCCCTACACTTCCT AATCGAATTAGAGGAAACCGCAATGGTAT TTTTAACACACAATCAATAATATTCTGTGGTT TTCAGCACCAAATACCAAAAGAAATAACC CCGAGTAAGCCAACGCCACAAACTGCA TCGAAATGAAGCTACTGTCTTCTAT-3'; 5'-TATGCTGTTTTGCTGAAATTGTTTTACTT AATCATATAACACCTTCCTCTATCGTCCTTT GTGTTGCTTGCTTCTCATGTTCATTATTATGTTG GAAATATTATTTAATAATATAGATTGTGTAA TTATCCATTTTGCGTTGTTGTTACACCCACCC TTTGCTGCTGCGC -3'.

The following primers were used to screen for potential correct knock-outs by amplifying a region within *rab7* or within the cassette:

rab7-KOscrn-Fwd AGAGGAAACCGCAATGGTAT; rab7-KOscrn-Rev TTCCTCTATCGTCCTTTGTG; GL5 (Cassette Fwd) GAAATCACGGCTAGTAAAATTG AT; RR1 (Cassette Rev) CTTAGCGACGTGTTCACT TTGCT.

PCR on the 12 lethal lines indicated presence of the knock-in cassette and absence of the *rab7* open reading frame.

## Genetics and fly culture

Eye mosaics were generated using the ey3.5FLP insertion on the X chromosome generated by Iris Salecker (*Chotard et al., 2005*; *Mehta et al., 2005*). The *rab7^Gal4-knock-in^* chromosome was recombined on an FRT82B chromosome using G418 selection. For rescue experiments UAS-venus-*rab7* variants were introduced on the second chromosome. Flies were kept at 25°C unless otherwise indicated. Dark-reared flies were kept in the same room next to light-raised flies. For constant light raising fly vials were kept in an aluminium foil covered box that provided even illumination provided by a Leica KL1500LCD cold light source. Light intensity was adjusted to 600 Lux using a hand-held digital lux meter at the level of the fly vials, 20 cm from the light source. To reduce expression levels of the UAS-*rab7* transgenes we generated flies of the genotype UAS-*rab7*-X/+; *rab7^Gal4-knock-in^*/tub-Gal80ts. The tub-Gal80ts stock was obtained from the Bloomington Stock Center (stock number 7018). To reduce level to endogenous levels flies were kept at 22°C. To reduce levels to 50% of endogenous levels, flies were kept at 18°C.

## Immunohistochemistry, imaging, and image processing

Larval and pupal brains and adult retinae were dissected and prepared for confocal microscopy as previously reported (*Williamson and Hiesinger, 2010*; *Williamson et al., 2010*). The tissues were fixed in phosphate buffered saline (PBS) with 3.5% formaldehyde for 15 min and washed in PBS with 0.4% Triton X-100. High-resolution light microscopy was performed using a Confocal Microscope (Leica SP5). Imaging data was processed and quantified using Amira 5.3 (Indeed, Berlin, Germany) and Adobe Photoshop CS6 as described (*Williamson et al., 2010*). The following antibodies were used: Chaoptin

(at 1:100), CSP (1:50), Hrs (at 1:300), Rab7 (at 1:500). Secondary antibodies used were Cy3 and Cy5 (Jackson ImmunoResearch Laboratories, Inc., West Grove, PA) raised against guinea pig, mouse or rabbit. Rhabdomeres were labeled with phalloidin (at 1:500) as previously reported (*Haberman et al., 2012*). Live imaging was performed according to published protocols (*Williamson and Hiesinger, 2010*) with the following modifications: Pupal eye-brain complexes were dissected in ice-cold Schneider's *Drosophila* Medium (Gibco, Grand Island, NY) at P+20–30%. Dissected tissues were pipetted into a drop of 0.4% dialyzed agarose (in Schneider's Medium) on a Sylgard layer. The sample was covered with a coverslip. Imaging was performed with a Leica SP5 resonant scanner, using a 63X glycerol lens. A time interval of 30 s was used for live imaging. The voxel size in all scans was 0.096 × 0.096 × 0.500 μm.

The compartment localization was quantified using Amira 5.3 (Indeed, Berlin, Germany) on high-resolution 3D confocal datasets. The confocal scans were obtained in 8 bit with no fluorescence signal outside the dynamics range. 10 cells were selected per genotype and individually cropped to yield separate datasets. Each dataset was separately binarized by threshold segmentation and visually controlled for only including clearly distinct compartments under exclusion of diffuse labeling. The cumulative fluorescence intensity of all voxels inside and outside the thresholded compartments was determined for the entire cell volume. Finally, the ratio of cumulative fluorescence inside/total fluorescence was calculated for each cell. Mean ratio, standard error (SEM) and p values were calculated for each genotype.

## Electron microscopy

To view the cell bodies and terminals of photoreceptors of *rab7* mutant and control ommatidia by transmission electron microscopy, flies were fixed in a modified Karnovsky glutaraldehyde and formaldehyde fixative, followed by osmium, then embedded in Epon, sectioned at 50 nm, all as previously reported (*Meinertzhagen, 1996*). Images were obtained using a Philips Tecnai 12 at 80 kV. Images were captured with a Gatan 832 Orius SC1000 CCD camera using Gatan DigitalMicrograph software.

## Electroretinograms (ERGs)

ERGs were performed as described in *Fabian-Fine et al. (2003)* with the following modifications: flies were fixed using Elmer's non-toxic Glue-All. We used 2M NaCl in the recording and reference electrodes. Electrode voltage was amplified by a Digidata 1440A, filtered through a Warner IE-210, and recorded using Clampex 10.1 by Axon Instruments. A post-recording filter was also provided by the Clampex software. Light stimulus was provided in 1 s pulses by a computer-controlled white LED system (Schott MC1500). All ERG recordings in *Figure 1* and *Figure 6* were performed in matched white- genetic backgrounds, which are significantly more sensitive to light stimulation than pigmented eyes; recordings in *Figures 3 and 4* were performed in flies with a single copy of mini-white and the same level of eye pigmentation. For quantification of depolarization and 'on' transients all experiment were carried out in duplicate or triplicate with at least 10 recording for each genotype and experimental condition.

## Electrophysiology at the larval neuromuscular junction

Third instar larvae were dissected and recorded from as described (*Imlach and McCabe, 2009*). The larvae were dissected in modified, ice-cold HL3.1 (*Stewart et al., 1994*), containing (in mM): Sucrose 115, NaCl 70, MgCl$_2$ 11, NaHCO$_3$ 10, KCl 5, Trehalose 5, HEPES 5, CaCl$_2$ 0.5, at pH 7.2. To avoid damaging the microscope optics, the filets were adhered to a SYLGARD 184 (Dow Corning, Midland, MI) substrate with a cyanoacrylate glue that polymerizes in water; the dissection pins were removed after glue polymerization, and the motor neurons were severed. Current clamp recordings were performed at room temperature in HL3.1. Recording microelectrodes were pulled to a sharp point with resistances between 20–40 MΩ. Only muscles 6, in segments 2–4, with membrane potentials between −65 and −60 mV were recorded. Voltage signals were amplified using a Multiclamp 700B amplifier (Molecular Devices, LLC., Sunnyvale, CA). Signals were digitized using a Digidata 1440 (Molecular Devices, LLC.), filtered at 2 kHz, and recorded using Clampex 10.2 software (Molecular Devices, LLC.). Miniature endplate potentials (mEPPs) were recorded for 60 s and analyzed using the 'Template search' feature of Clampfit 10.2 (Molecular Devices, LLC.). Frequency was calculated by taking the number of events in a recording and diving by 60 s. For EPPs, stimulation was applied via the segmental motor nerve axons with a A365 stimulus isolator (WPI), under digital control of the Clampex software. Stimulus electrodes had a ~10 μm inner diameter after firepolishing. Superthreshold stimulation (between 3–4 mA, for

100 ms) was applied at 0.2 Hz for 60 s and recorded with Clampex. EPP amplitudes were measured using the Clampfit cursor function. Significance was calculated using Welch's *t* test.

## Protein extracts and Western blot analysis

Total proteins were extracted from fly heads in buffer containing 20 mM Tris, 150 mM NaCl, 1 mM PMSF, and 1x complete protease inhibitors (Roche) at pH 7.4. The fly head extract was mixed well in 1% Triton X-100 (BioRad, Hercules, CA) and incubated for 1 hr at 4°C. Samples were centrifuged at 16,000 × *g*, 15 min at 4°C to remove cell debris. The resulting supernatants were loaded on 12% SDS-PAGE and transferred to PVDF membrane. Primary antibodies used were: mouse anti-human Rab7 (1:2500); rabbit anti-fly Rab7 (1:5000); muse ant-actin (1:5000). Corresponding secondary antibodies were used at 1:10,000. The signals were detected with the *Pierce ECL* Western blotting substrate (Thermo Scientific, Rockford, IL). The quantification of Rab7 western blots was performed with ImageJ software (NIH, Bethesda, MD). Data were analyzed with GraphPad *Prism 4.*

## Acknowledgements

We would like to thank Konrad Basler, Patrick Dolph, Sean Sweeney, Koen Venken, Hugo Bellen, Adrian Rothenfluh, Matt Scott, the Bloomington Stock Center and the University of Iowa Developmental Studies Hybridoma Bank for reagents. We further thank Drs Nevine Shalaby, Michael Buszczak, Ilya Bezprozvanny, Soumik BasuRay and Joachim Herz and all members of the Hiesinger lab for discussion and critical comments on this manuscript. PRH is a Eugene McDermott Scholar in Biomedical Research.

## Additional information

### Funding

| Funder | Grant reference number | Author |
| --- | --- | --- |
| National Institutes of Health | RO1EY023333 | P Robin Hiesinger |
| Welch Foundation | I-1657 | P Robin Hiesinger |
| Muscular Dystrophy Association | MDA275948 | P Robin Hiesinger |
| National Institutes of Health | RO1EY03592 | Ian A Meinertzhagen |
| National Science Council of Taiwan | NSC 101-2320-B-002-051-MY2 | Chih-Chiang Chan |
| National Taiwan University College of Medicine | NTUMC, 102R39012 | Chih-Chiang Chan |
| National Institutes of Health | T32 NS069562 | Eugene Jennifer Jin |

The funders had no role in study design, data collection and interpretation, or the decision to submit the work for publication.

### Author contributions

SC, EJJ, Acquisition of data, Analysis and interpretation of data, Drafting or Revising the article; MNÖ, ZL, EA, DW, DE, Acquisition of data, Analysis and interpretation of data; W-HJ, Analysis and interpretation of data, Drafting or revising the article; IAM, Conception and design, Analysis and interpretation of data, Drafting or revising the article; C-CC, PRH, Conception and design, Acquisition of data, Analysis and interpretation of data, Drafting or revising the article

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
