## [Decision Letter]

Thank you for sending your work entitled “Charcot-Marie-Tooth 2B mutations in *rab7* cause dosage-dependent neurodegeneration due to partial loss of function” for consideration at *eLife*. Your article has been favorably evaluated by a Senior editor and 3 reviewers, of whom, Mani Ramaswami, is a member of our Board of Reviewing Editors.

The Reviewing editor and the other reviewers discussed their comments before we reached this decision, and the Reviewing editor has assembled the following comments to help you prepare a revised submission.

The work provides very interesting and compelling studies based on *rab7* loss of function mutations in *Drosophila* to challenge the current dogma that CMT2B-causative mutations found in human patients represent toxic gain of function mutations.

Through an in-depth characterization of transgenes carrying mutations in CMT2B causative mutations in *rab7*, the authors show that all 4 CMT2B causative mutations partially inactivate the protein, leading to the idea that the dominant inheritance pattern is due to dosage effects. The work creates a very useful in vivo model for CMT2B that can be used for further genetic and pharmacological analysis. The study is comprehensive, generates new valuable reagents, carries out multiple functional analyses (activity-dependent neurodegeneration, loss of synaptic transmission) and tries to get at the mechanism by which a loss of function in the mutant rab7 protein might result in progressive degeneration in sensory neurons in the eye. The data are generally nicely presented, though proper labels are needed in all of the figures.

To establish the haploinsufficiency of the human disease mutations the authors use a series of well-designed and very well executed experiments, including an elegant experiment in quantitative genetics to show that: (a) Human *rab7* rescues loss of fly *rab7*; (b) Fly *rab7* is haploinsufficient when it comes to neuroprotection, as loss of one copy causes activity dependent degeneration; and (c) both the fly and human disease variants are not toxic when overexpressed at reasonable levels in vivo. Interestingly, this is in fact consistent with the majority of biochemical analyses of the human CMT2B mutant *rab7*'s.

Substantive concern: What the paper lacks is clear cut experimental evidence showing that a single copy of a CMT2B *rab7* variant (or protein expression equivalent of it) in a heterozygous mutant background represents a reduction in Rab7 activity and predisposes neurons to degeneration. The authors should attempt this experiment in any way they chose. The simplest would appear to be to use the *rab7*-*gal4* knock-in heterozygotes and homozygotes to express the different CMT2B variants at levels approaching 50% of the endogenous protein: e.g., by exploiting the temperature sensitivity of the Gal4/UAS system, and testing if the resulting flies (which most closely model CMT2B patients) show neurodegeneration, particularly in older flies. The result should be included in this paper, although the reviewers acknowledge that there could be limitations of the fly system that complicate one's ability to model the disease with such extreme precision (that is, a negative result, clearly discussed, will also satisfactorily complete this otherwise comprehensive study).

---

## [Author Response]

*What the paper lacks is clear cut experimental evidence showing that a single copy of a CMT2B* rab7 *variant (or protein expression equivalent of it) in a heterozygous mutant background represents a reduction in Rab7 activity and predisposes neurons to degeneration. The authors should attempt this experiment in any way they chose. The simplest would appear to be to use the* rab7*-*gal4 *knock-in heterozygotes and homozygotes to express the different CMT2B variants at levels approaching 50% of the endogenous protein: e.g., by exploiting the temperature sensitivity of the Gal4/UAS system, and testing if the resulting flies (which most closely model CMT2B patients) show neurodegeneration, particularly in older flies. The result should be included in this paper, although the reviewers acknowledge that there could be limitations of the fly system that complicate one's ability to model the disease with such extreme precision (that is, a negative result, clearly discussed, will also satisfactorily complete this otherwise comprehensive study)*.

We have performed the suggested experiments. In initial experiments, we only succeeded in reducing the expression levels of the disease mutant proteins to about 2-fold, despite the temperature sensitivity of the Gal4/UAS system. To obtain flies with endogenous levels of disease mutant protein expression, we performed a series of experiments introducing Gal80ts to further reduce Gal4/UAS expression of the CMT2B variants. These experiments were facilitated by the fact that the disease mutant proteins are fluorescently tagged, allowing quantitative comparison with the endogenous protein in Rab7 Western Blot. As shown in the new Figure 7, we identified the conditions to generate flies that express the human disease mutant variants at levels comparable to heterozygous endogenous levels, as well as ∼50% below endogenous levels, and up to more than 10-fold endogenous levels.

Light stimulation causes degeneration in *rab7* heterozygotes, which can be rescued by expression of wild-type Rab7. Our new data reveal that endogenous levels of the CMT2B variants and even 50% below endogenous levels are sufficient to rescue this phenotype (new Figure 7). We conclude that we cannot easily mimic the precise dosage-dependence of the human condition in flies. This is consistent with our previous conclusions, as shown in the model figure (now Figure 8): We proposed that putative human heterozygotes, which have never been documented, are likely not viable, whereas the fly heterozygotes clearly are almost completely healthy. Adding 10-50% of a second *rab7* copy, as is the case with the CMT2B variants, permits normal human development and function for years, but is not enough to prevent slow degeneration after 12 or more years as observed in patients. In contrast, in the fly the addition of 10-50% of *rab7* function in the form of a CMT2B allele is sufficient to quantitatively rescue the degeneration phenotype within the limits our functional and morphology read-outs.

We conclude that the slow, dosage-dependent effect of a mild reduction of Rab7 function in patients that only leads to degeneration after 12-40 years is not easily modeled in *Drosophila*. This is a clear limit of the fly model. However, we would like to note that mimicking slow, adult-onset degeneration as it occurs in patients over many years is a problem in all animal models, including mouse models for numerous neurodegenerative diseases. Even a mouse model for mildly reduced Rab7 function may not precisely mimic the dosage-dependence and slow disease progression observed in human patients, and in cell culture equivalent experiments are impossible. We discuss this and other limits of the *Drosophila* model now in the revised Results as well as an extra Discussion section entitled ‘Limitations of the *Drosophila* model’.

Nonetheless, we still consider the experiment requested by the reviewers very valuable. Not a single one of the numerous previous cell culture overexpression papers on CMT2B provided any information on the level of CMT2B protein overexpression used in these studies. Since we can precisely quantify the level of expression of the CMT2B variants (in the endogenous expression pattern in an entire organism in vivo), we have also repeated the rescue/overexpression experiment with the CMT2B proteins with more than 10-fold overexpression. As shown in the new Figure 7, even more than 10-fold overexpression rescues and does not negatively affect neuronal function.

Taken together, we can make two strong and quantitative statements after these revision experiments. First, Rab7 CMT2B proteins at 50% less than endogenous heterozygous levels (corresponding to 25% of total Rab7 function) retain sufficient wild-type function to rescue rab7-dependent degeneration. Second, Rab7 CMT2B proteins at >10-fold overexpression rescue equally well and exhibit no toxic or other obvious gain-of-function effects. Together with our quantitative colocalization and live imaging experiments that show reduced function at 10-50% wild type levels, these data corroborate our conclusion that CMT2B proteins represent partial loss-of-function alleles with no toxic gain-of-function in *Drosophila*.